# Biodegradable Mulch Films and Bioformulations Based on *Trichoderma* sp. and Seaweed Extract Differentially Affect the Metabolome of Industrial Tomato Plants

**DOI:** 10.3390/jof10020097

**Published:** 2024-01-25

**Authors:** Alessia Staropoli, Ida Di Mola, Lucia Ottaiano, Eugenio Cozzolino, Angela Pironti, Nadia Lombardi, Bruno Nanni, Mauro Mori, Francesco Vinale, Sheridan Lois Woo, Roberta Marra

**Affiliations:** 1Department of Agricultural Sciences, University of Naples Federico II, Piazza Carlo di Borbone, 1, 80055 Naples, Italy; alessia.staropoli@unina.it (A.S.); ida.dimola@unina.it (I.D.M.); lucia.ottaiano@unina.it (L.O.); angela.pironti@unina.it (A.P.); nadia.lombardi@unina.it (N.L.); bruno.nanni@unina.it (B.N.); mori@unina.it (M.M.); 2Institute for Sustainable Plant Protection, National Research Council, Piazzale Enrico Fermi, 1, 80055 Naples, Italy; frvinale@unina.it; 3Council for Agricultural Research and Economics, Research Center for Cereal and Industrial Crops, Viale Douhet, 8, 81100 Caserta, Italy; eugenio.cozzolino@crea.gov.it; 4BAT Center—Interuniversity Center for Studies on Bioinspired Agro-Environmental Technology, University of Naples Federico II, Piazza Carlo di Borbone, 1, 80055 Naples, Italy; woo@unina.it; 5Department of Veterinary Medicine and Animal Production, University of Naples “Federico II”, Via Delpino, 1, 80137 Naples, Italy; 6Department of Pharmacy, University of Naples Federico II, Via Domenico Montesano 49, 80131 Naples, Italy

**Keywords:** *Solanum lycopersicum*, *Trichoderma*, seaweed extract, biostimulant, biodegradable film, metabolomic analysis, alkaloids, flavonoids

## Abstract

The use of biostimulants and biofilms in agriculture is constantly increasing, as they may support plant growth and productivity by improving nutrient absorption, increasing stress resilience and providing sustainable alternatives to chemical management practices. In this work, two commercial products based on *Trichoderma afroharzianum* strain T22 (Trianum P^®^) and a seaweed extract from *Ascophyllum nodosum* (Phylgreen^®^) were tested on industrial tomato plants (*Solanum lycopersicum* var. Heinz 5108F1) in a field experiment. The effects of single and combined applications of microbial and plant biostimulants on plants grown on two different biodegradable mulch films were evaluated in terms of changes in the metabolic profiles of leaves and berries. Untargeted metabolomics analysis by LC-MS Q-TOF revealed the presence of several significantly accumulated compounds, depending on the biostimulant treatment, the mulch biofilm and the tissue examined. Among the differential compounds identified, some metabolites, belonging to alkaloids, flavonoids and their derivatives, were more abundant in tomato berries and leaves upon application of *Trichoderma*-based product. Interestingly, the biostimulants, when applied alone, similarly affected the plant metabolome compared to control or combined treatments, while significant differences were observed according to the mulch biofilm applied.

## 1. Introduction

Tomato (*Solanum lycopersicum* L.) represents the most economically important vegetable crop in the world, with approximately 34.8 million tons of product used in the processing industry [1]. For industrial tomatoes, producers prefer varieties that guarantee high and constant production, adaptability in different soil situations, resistance to the most important and widespread diseases, and suitability for mechanical harvesting. In addition to the valuable organoleptic characteristics, tomato fruits have several health properties: they are rich in fibre, mineral salts, vitamins C and E, secondary metabolites such as carotenoids (lycopene and β-carotene) and phenolic compounds (flavonoids) and all antioxidants with well-known beneficial effects against cardiovascular diseases, tumour growth and diseases related to ageing [2]. Until a few years ago, the sole parameter producers considered was the yield. However, today, we are increasingly moving towards long-term sustainable agriculture, where various crucial parameters come into play. These include dry matter content, acidity levels, and Brix degrees, among others.

Possible solutions to increase agricultural production, meet food needs and reduce environmental impact include the use of biostimulants, products that include beneficial microorganisms (such as arbuscular mycorrhizal fungi and *Trichoderma* spp.) and natural substances (humic acids, algae and plant extracts, protein hydrolysate and silicon) capable of stimulating vigour, growth and yield of crops, even in non-optimal conditions [3]. Biostimulants are able to activate different physiological and biochemical processes that lead to an increase in the efficiency of water and nutrient use [4,5,6]. They represent a valid alternative to chemicals as they do not pose threats to biodiversity, nor do they produce harmful effects on human health and the environment. Moreover, they aid in diminishing reliance on synthetic fertilizers and toxic pesticides [7].

Plant biostimulants are derived from organic fresh substances that contain a rich variety of bioactive compounds. These biostimulants primarily consist of minerals, humic compounds, vitamins, chitin/chitosan, amino acids, and poly- and oligosaccharides [5]. Among plant biostimulants, brown algae extracts are products usually deriving from the extraction of algae biomass of the genera *Ascophyllum*, *Ecklonia*, *Macrocystis* and *Durvillea*. The method of extraction and the temperature at which this process takes place lead to a differentiation of the final product. However, there are some fundamental qualities related to seaweed extracts that can be responsible for the biostimulant activity and are common to the majority of products: these are plant hormones such as abscisic acid (ABA), gibberellic acids (GAs), auxins, brassinosteroids and cytokinins (CK), and growth regulators such as betaines and algal polymers, especially polysaccharides such as alginates, fucoidans, mannitol and laminarin [8].

The beneficial microorganisms that can be used as biostimulants include fungi of the genus *Trichoderma* (fam. Hypocreaceae). Since 1930, the ability of *Trichoderma* spp. to act as a biocontrol agent was demonstrated [9]; successively, numerous studies have consistently confirmed their ability to control phytopathogens as well as to establish beneficial interactions with host plants, resulting in increased availability of nutrients, promotion of plant growth, mitigation of the effects of abiotic stress and induction of defence mechanisms [10]. These characteristics, as well as the ease of storage and the abundant production of conidia, make *Trichoderma* one of the most used microorganisms in agriculture. Indeed, in recent years, the number of products containing this fungus has increased exponentially, with up to 144 registrations in 40 countries [10]. In the context of commercial *Trichoderma*-based bioformulations, one of the most widely used isolates is the T22 strain of *T. afroharzianum* (Rifai strain KRL-AG2, ex *T. harzianum*) [11]. This strain protects the roots of plants from the attack of various telluric pathogens, including *Fusarium* spp., *Pythium* spp., *Rhizoctonia* spp. and *Sclerotinia* spp., and is capable of inducing systemic resistance (ISR) in several plants of agricultural importance, including tomato [12,13,14]. It can be used for preventive purposes on a large number of crops, in the case of some foliar diseases, especially when complete resistance is required. It is particularly useful as an alternative when large doses of fungicides are needed. Finally, T22 exhibited efficacy in specifically preventing viral diseases and certain bacterial diseases, making it a valuable asset in agricultural practices, specifically preventing viral diseases and some bacterial diseases.

In this study, we evaluated the effects on industrial tomato plants (*Solanum lycopersicum* L. var. Heinz 5108) resulting from the application of two commercial bioformulations based on a seaweed extract or *Trichoderma* T22, respectively, used in combination with different biodegradable mulching films. Both quantitative and qualitative parameters on plants grown in field conditions were previously examined by Di Mola et al. [15]. Here, a metabolomic analysis was conducted on leaves and fruits to discriminate the effects of field treatments in terms of accumulated differential metabolites. The results presented here and in the previous work overall show the influence of plant biostimulants and biodegradable mulch films on field-grown tomato plants, thus highlighting their potential applications in sustainable agriculture. However, metabolomic analysis provides a broader picture of the treated plant responses reflected in the biosynthetic pathways, i.e., the cause and effects of cultivation practices on vegetative tissues.

## 2. Materials and Methods

### 2.1. Plant Material and Pre-Transplant Processing

During the 2021 spring–summer season, tomato seedlings (*Solanum lycopersicum* L. variety Heinz 5108 F1′, Agrisem srl Soc., San Valentino Torio, Italy) were transplanted in a field located in Frignano (CE), southern Italy (latitude: 41.02, longitude: 14.17), with a density of 0.12 pt m^−2^ in paired rows at 50 cm distance. The area is characterized by a temperate Mediterranean climate with average annual temperatures of 15.2 °C and average annual rainfall of 900 mm concentrated in the autumn–winter period, while summers are dry. Before the transplant, crop residues were shredded and a fertilizer, TMAC Sprint (title NPK 10-5-12), was spread at a dose of 700 kg ha^−1^. Tilling was carried out 20 days before the transplant to simulate a false sowing. Upon transplant, milling was performed to further refine the soil.

Two different biodegradable mulch films were applied to the soil: a starch-based material (Mater-Bi^®^, Novamont S.p.A., Novara, Italy) and a compostable polymer composed of ecoflex^®^, poly-lactic acid and other additives (Ecovio^®^, BASF, Mannheim, Germany). Both films were 15 µm thick. The films were hand-placed after hoses for drip irrigation were rolled out.

### 2.2. Biostimulants

In this work, two commercial biostimulants were used: (i)Phylgreen^®^ (Trade Corporation International, Madrid, Spain), a liquid formulation consisting of the pure extract of the seaweed *Ascophyllum nodosum*. The formulation is soluble and has a high content of alginates, vitamins, natural antioxidants and noble amino acids; it is used to improve flowering and fruit setting, acts as a promoter of photosynthesis and root development, and increases plant tolerance to environmental stress.(ii)Trianum-P^®^ (Koppert B. V., Berkel en Rodenrijs, The Netherlands), containing the microorganism *Trichoderma afroharzianum* (ex *T. harzianum*) Rifai strain KRL-AG2 (T-22). The bioformulation contains a minimum concentration of 1 × 10^9^ CFU g^−1^. It has been registered as a Plant Protection Product (PPP), but it is also able to stimulate plant growth and development.

Biostimulants were applied singly or in combination, as described in the following paragraph.

### 2.3. Experimental Design and Field Management

The experiment was organized according to a factorial completely randomized design with two factors, biostimulants (two biostimulants and one untreated control) and mulch biofilms (two biodegradable films and one bare soil as control). The field trial was divided into 3 plots with 3 replicates, each measuring 14 m × 0.7 m, and spaced 1 m apart; each plot was mulched and divided into 4 sub-plots of 3 m^2^, each consisting of 10 plants (Appendix A). The sub-plots received biostimulants applications according to the following scheme: (i) Phylgreen^®^ alone (BIO); (ii) Trianum-P^®^ alone (MICRO); (iii) combination of Phylgreen^®^ and Trianum-P^®^ (MICRO+BIO); (iv) no biostimulants (water control).

After transplant, the microbial biostimulant was applied by irrigating each plant with 50 mL of a suspension containing 2 × 10^6^ CFU mL^−1^. Treatments were repeated monthly with 100 mL plant^−1^, for a total of 4 applications. Plant-based biostimulant was applied as a foliar spray at a rate of 3 mL L^−1^ on a bi-weekly basis. Control plants were sprayed with tap water.

### 2.4. Plant Sampling

Both leaf and fruit samples were collected at harvesting. About 10 leaves and 10 berries were sampled for each replicate of each treatment. Aliquots of each sample were frozen in liquid nitrogen, freeze-dried, and subjected to metabolomic analyses. 

### 2.5. Metabolites Extraction

The extraction of metabolites from tomato leaves and berries was performed using the protocol described by Staropoli et al. [16]. The freeze-dried leaf and berry samples were ground. Subsequently, a 100 mg aliquot was taken from each of them to which 2 mL of LC-MS-grade (liquid chromatography-mass spectrometry) methanol (MeOH) was added. The samples were then vortexed for 1 min and stored for 1 h at a temperature of 4 °C. Subsequently, the mixture was centrifuged for 20 min at 12,000 rpm and 4 °C. The supernatant was then recovered and used for metabolomic analyses.

### 2.6. LC-MS Analysis

Analyses were carried out on an Agilent HP 1260 Infinity series liquid chromatograph coupled to a Q-TOF (quadrupole-time of flight) mass spectrometer and equipped with a DAD system (diode array, Agilent Technologies, Santa Clara, CA, USA). An Adamas^®^ C-18-C-Bond column (4.6 × 50 mm, 3.5 µm, SepaChrom, Rho, Milan, Italy) was used for the chromatographic separation and held at a constant temperature of 37 °C. Analyses were performed at a flow rate of 0.5 mL min^−1^ using a linear gradient system composed of 0.1% (*v*/*v*) formic acid in water (eluent A) and 0.1% (*v*/*v*) formic acid in acetonitrile (eluent B). The elution gradient was set as follows: 5% to 70% eluent B in 4 min, isocratic 70% eluent B in 4 to 5 min; 70% to 80% eluent B from 5 to 8 min and 80% to 100% eluent B for 8 to 10 min; 100% to 50% eluent B from 10 to 12 min and finally, lowering to starting conditions (5% eluent B) from 12 to 14 min. After returning to initial conditions, equilibrium was reached after 1 min. The injection volume was 10 µL. UV spectra were collected by the DAD detector every 0.4 s from 190 to 750 nm, with a resolution of 2 nm. The MS system was equipped with a dual electrospray (ESI) ionization source operating in both positive and negative modes. The capillary was held at 2000 V, fragmentor voltage at 180 V, cone 1 (skimmer 1) at 45 V and Oct RFV at 750 V. The gas flow rate was set at 11 L min^−1^, at 350 °C and the nebulizer was set at 45 psig. Mass spectra were recorded in the *m*/*z* range 100–1700 as centroid spectra, with three scans per second. To perform real-time mass correction, a solution consisting of purine (C_5_H_4_N_4_, *m*/*z* 121.050873, 10 µmol L^−1^), and hexakis (1H,1H,3H-tetrafluoropentoxy)-phosphazene (C_18_H_18_O_6_N_3_P_3_F_24_, *m*/*z* 922.009798, 2 µmol L^−1^) was constantly infused by an isocratic pump (1260 Infinity series, Agilent Technologies) with a flow rate of 0.05 mL min^−1^. All MS and HPLC parameters were set up with Agilent MassHunter data acquisition software, version B.05.01 (Agilent Technologies).

### 2.7. Statistical Analysis

Statistical analysis was performed using the Mass Profile Professional software, version 13.1.1 (Agilent Technologies). Raw data came from leaf and berry extracts. Normalization and alignment parameters were as follows: abundance filter, >5000 counts; minimum number of ions, 2; alignment RT window, 0.4 min intercept, and 0% slope; alignment mass window, 2 mDa intercept, and 20 ppm slope. The normalized features were filtered again, and only masses appearing at least in two of three samples were accepted. Background noise was removed by subtracting masses found in blank runs from filtered masses. The extracted ion chromatogram (EIC) of each endogenous metabolite was extracted with ±20 ppm single ion expansion using the Mass Hunter software v B.06.00. Normalized data were pooled by field-applied treatment (i.e., individual biostimulants and mulches), and these groups were subjected to one-way ANOVA analysis with Tukey’s post hoc test (*p*-value < 0.05) and Benjamini–Hochberg procedure for *p*-value correction; finally, a fold change ≥ 2.0 was applied. Then, the results obtained were subjected to hierarchical clustering (Euclidean similarity measure and Wardsa linkage rule applied) in order to compare the metabolic profiles of the plants and detect the differences induced by the different treatments and mulches.

Statistically relevant compounds were tentatively identified using an in-house database comprising data from the METLIN library and from the literature, with a mass accuracy of 10 ppm. Empirical formulas were generated for unknown compounds with the following parameters: ppm limit = 10, isotope model = common organic molecules, limit charge state to a maximum of 2, and use +H or −H, or sodium and potassium adducts

Abundance values for each condition of identified compounds were used to draw two heatmaps (one for leaves and one for berries) using the pheatmap package in R (version 4.2.2, R Core Team, Vienna, Austria, 2020).

## 3. Results

### 3.1. Metabolites Identification

At harvesting, samples of tomato leaves and berries were collected, and metabolites were extracted and analyzed by LC-MS, to evaluate the effect of field treatments on the plant chemical profile. Chromatograms and mass spectra were compared with data reported in literature and freely available databases. Putatively identified compounds in leaves and berries are reported in Table 1, including retention time, experimental and theoretical monoisotopic mass, molecular formula, and type of tomato tissue (leaf or berry). Results showed the presence of 65 compounds, of which the main ones are flavonoids and derivatives, phenols, alkaloids and derivatives, saponins, fatty acids, terpenes, amino acids and their derivatives (Table 1).

In this work, the production of compounds in tomato leaves and berries was correlated with the application of biostimulants combined with two different types of mulch. The metabolomic analysis was initially conducted with an untargeted approach, i.e., not linked to the identification of a specific metabolite or a class of compounds. For this type of analysis, the differences found in the metabolome of the leaves/berries of plants treated with the two biostimulants compared to plants treated with water (control) were evaluated. Moreover, differences between plants treated with the microorganism (applied alone or in combination with the plant biostimulant) with either the two types of mulch biofilms (Ecovio^®^ and MaterBi^®^), and those grown on bare soil (control) were examined. Subsequently, the most representative compounds were chosen and analyzed through a targeted approach.

### 3.2. Untargeted Metabolomic Analysis

Untargeted metabolomic analysis provided insights into the changes that occurred in the metabolic profile of the extracts due to the treatments over time. One-way ANOVA statistical analysis was carried out, which highlighted the presence of about 300 compounds whose accumulation was statistically different (*p* < 0.05) both in the berries and leaves. Several compounds showed a significant and consistent change (fold change ≥ 2.0) after the application of T22 and/or Phylgreen^®^, as reported in the following paragraphs. To visually represent the differences in relative abundance across the tissues, heat maps were constructed using peak intensities obtained from the LC-MS analysis. 

#### 3.2.1. Leaves

The effect of biostimulant treatments in terms of production of differential metabolites in tomato leaves is reported in Table 2, which indicates the total number of compounds analyzed in positive (ESI+) and negative (ESI−) ionization modes. Compared to the control (plants treated with water only), a greater number of differential metabolites whose intensity increased (UP) was observed in plants treated with the plant biostimulant. On the other hand, in the presence of *Trichoderma* (with or without Phylgreen^®^), more metabolites with lower intensity (DOWN) were observed. Moreover, the use of mulch biofilms and biostimulants affected tomato leaf metabolome; for both types of films, a greater number of differential metabolites, whose intensity decreased compared to control (plants grown on bare soil and treated with *Trichoderma*), was observed (Table 2). Finally, the effect of the mulch films and the combined biostimulant (T22 and Phylgreen^®^) treatments compared to the control (plants grown on bare ground and treated with the combination of the two biostimulants) were investigated. For both Ecovio^®^ and MaterBi^®^ biofilms, a greater accumulation of up-regulated metabolites, compared to control, was observed (Table 2).

A grouping of samples was then carried out based on the abundance of continuous variables (Hierarchical clustering) by joining the replicates of the treatments. As already emerged from the previous tables, most of the significant differentially accumulated metabolites are the ones whose normalized intensity values of abundance are <0 (highlighted in blue). In general, the results obtained show a different metabolomic profile in treated plants compared to controls. Hierarchical clustering, showing the effects of single and combined treatments on plants cultivated on bare soils, for both ionization modes (ESI+ or ESI−), are reported in Figure 1. For positive ionization mode, individually applied biostimulants (Bio and Micro) affected leaves metabolic profiles in a similar way, determining a different metabolomic profile compared to both the combined treatment (T22 and Phylgreen^®^, MicroBio) and the control group (Figure 1A). In negative ionization mode, on the other hand, Phylgreen^®^-based treatments (Bio and Micro+Bio) clustered together and similarly influenced leaves metabolite profiles (Figure 1B).

The effect of the two different types of mulch on the metabolome of plants treated with T22 is reported in Figure 2. The two pictures show opposite situations: while in ESI+ mode the metabolites that have a high relative abundance predominate (especially in the samples with Materbi^®^ mulch film and in the bare control, Figure 2A), in ESI− mode differential metabolites are mainly present with a lower abundance (Figure 2B). However, Ecovio^®^ (Eco) mulching film determined the greatest differences in the metabolomic profile of the leaves.

Finally, comparing the metabolomic profile of the leaves obtained following the combined application of the microbial and vegetable biostimulant with the two different mulches (Eco and Nov) with respect to the control (bare soil treated with the combination of biostimulants), we obtain both positive (ESI+) and negative (ESI−) results similar to those obtained with T22 alone. As shown in Figure 3, in the positive ionization, the Ecovio^®^ mulch biofilm (Eco) showed greater differences in the plant metabolomic profile compared to MaterBi^®^ and the control (Figure 3A); in the negative ionization mode, the two mulches give similar and different results with respect to the bare soil (Figure 3B).

#### 3.2.2. Berries

The effect of biostimulant treatments on the production of metabolites in tomato berries is shown in Table 3, which indicates the total number of compounds analysed in positive (ESI+) and negative (ESI−) ionization modes. Compared to the control (plants treated with water only), a greater number of differential metabolites was observed whose intensity decreased (DOWN) in all conditions, especially when the two biostimulants were applied in combination. Table 3 also shows the effect of the different mulch biofilms and the treatment with the microbial biostimulant T22 on the metabolome of tomato berries, compared to the control (plants grown on bare soil and treated with *Trichoderma*). In this case, for both mulches, a greater number of differential metabolites was observed whose intensity increased (UP), particularly in the berries of plants grown on MaterBi^®^ mulch (Nov). Finally, the effect of the mulch biofilms on the combined treatment of the two biostimulants compared to the control (plants grown on bare soil and treated with biostimulants) was analysed. While with Ecovio^®^, 52 DOWN and 25 UP differential metabolites were produced, with MaterBi^®^ mulch film (Nov), the number of differential metabolites whose intensity increased or decreased compared to control was similar (38 UP and 39 DOWN; Table 3).

Also, for the berries, a grouping of the samples based on the abundance of continuous variables (Hierarchical clustering) was carried out and heat maps were created as previously conducted for the leaf extracts. Appendix A shows the groupings of differential metabolites obtained in positive (ESI+; Appendix A) and negative (ESI−; Appendix A) ionization modes when only treatments with biostimulants applied individually or in combination were taken into account.

Comparing the metabolomic profile of the three treatments (biostimulants used singly or in combination) applied on plants grown on bare soil, compounds with lower relative abundances predominate. In both analyses, the single application of biostimulants resulted in similar variations in the metabolome of the berries. 

However, a different situation was observed when the metabolomic profiles of fruits produced by plants treated with T22 but grown on mulched soils were compared (Appendix A). In ESI+ mode, a similar metabolomic profile in T22-treated plants grown on mulched biofilms was observed compared to the controls (inoculated plants grown on bare soil; Appendix A). Conversely, in negative ionization (ESI−) mode, control samples and plants grown onEcovio^®^ mulching film showed a similar metabolomic profile, while in the presence of MaterBi^®^, a greater number of differential metabolites with high relative abundance was found (Appendix A).

Finally, the changes in the differential metabolites accumulated in tomato berries with the combined treatments T22+ Phylgreen^®^ were compared in relation to the two mulch films used (Appendix A). The plants grown on bare soil and treated with the same combination of biostimulants represent the control (Bare Micro+Bio). In this case, in addition to a smaller number of differential metabolites, it was observed in both ionization modes that with the Ecovio^®^ mulch, there was a metabolomic profile more similar to the control, while with the MaterBi^®^, greater differences were found, mainly consisting of metabolites with lower relative abundance in ESI+ (Appendix A) and metabolites with relative abundance ≥0 in ESI− (Appendix A).

### 3.3. Targeted Metabolomic Analysis

Significant differential metabolites were putatively identified and their abundance was evaluated as a function of the biostimulant-based treatments and the presence of mulch biofilm compared to plants grown on bare soil (Appendix A). The abundance of the main differential metabolites accumulated in treated tomato plants (leaves and berries), grown on mulched or bare soils and compared to the controls (plants grown on bare soil and treated with water only BARE CTRL, or plants grown on bare soil and treated with the *Trichoderma*-based biostimulant-BARE MICRO) is reported in Table 4. Interestingly, contrary to the trend observed in leaves, tomato berries showed a significant decrease in the abundance of differential metabolites compared to control samples.

Moreover, for some compounds, a significant accumulation in treated samples compared to control groups was observed, depending upon the treatment applied and on the tissue/plant organ examined (Table 4). For instance, dehydrotomatine was more abundant in berries treated with biostimulants, whereas it was less abundant in leaves treated with T22 compared to the control; α- and β1- tomatine were identified as differential metabolites in leaves only, except for the comparison between berries grown in mulched soil (Ecomont^®^) and treated with T22 or in the control (Table 4). However, in most cases, statistically different compounds were identified only in one of the two types of samples examined. The classes of putatively identified compounds were flavonoids, phenols, alkaloids, saponins, fatty acids, terpenes and amino acids, amongst others (Table 1). Among these, the flavonoids and alkaloids are the most represented and showed a uniform trend in the tissues examined. The abundance of numerous flavonoids and derivatives identified in leaves strongly increased in treated plants; conversely, the abundance of alkaloids and derivatives was less influenced by treatments. Among all significant metabolites, three were chosen as being discriminating by abundance and findings in the literature. Within the flavonoid group, (+)-medicarpin showed a strong increase in abundance in leaves after the biostimulant treatments. Particularly, single treatments with Phylgreen^®^ or with T22 induced an increase in the intensity of 60,000 and 80,000 times, respectively.

Interestingly, among alkaloids, α-tomatine was detected in both tomato leaves and berries (Table 1), but its abundance decreased in both mulched-treated plants following T22 inoculation (Table 4). Conversely, tomatidine and dehydrotomatine showed higher abundance in tomato berries following inoculations with T22 compared to leaves, both in plants grown in bare soil and using mulching biofilms (Table 4).

## 4. Discussion

The use of mulch films typically results in increased production yields, attributed to several factors, such as increased soil temperature, preserved soil humidity, weed and pest reduction, and more efficient soil nutrient utilization [17,18]. While black polyethylene plastic films are commonly used in agriculture, there is a growing consideration for biodegradable plastic films due to concerns over traditional film disposal after crop harvesting. The conventional removal and disposal process can lead to economic and environmental problems, including recovery and disposal costs, as well as the generation of toxic substances through film incineration [19]. In addition to environmental protection, several studies support the performance of biodegradable films in terms of the quality and yield of vegetable crops [20,21].

The application of microorganisms and biostimulants is becoming increasingly popular in agriculture, mainly because of their positive effects on plant disease control, growth stimulation, nutrition, and the production of beneficial bioactive secondary metabolites (SMs). Fungi from the genus *Trichoderma*, initially studied as effective mycoparasites, are employed as microbial biostimulants because of their ability to act as plant growth promoters (PGPs) on various crops, thus leading to enhanced root development, increased leaf surface area, higher production yields, improved nutrient content and induction of systemic resistance [10,22,23,24]. Recent studies have also placed emphasis on the involvement of secondary metabolites produced by these beneficial fungi in interactions with plants [25].

Seaweed extracts, especially those derived from brown algae like *Ascophyllum nodosum* (L.), are among the most widely used biostimulants in agriculture [5,8,26]. The application of these extracts has been found to promote the growth, yield, quantity, and quality of various crops [27]. In a study conducted by Fleming et al. [28] using Phylgreen^®^, it was demonstrated that this *A. nodosum*-based product had a significant plant growth promotion effect on *Arabidopsis thaliana* in the following 7 days from the first treatment even in severe drought conditions.

In this study, the effects of the application of the commercial strain T22 of *T. afroharzianum* (Trianum-P^®^) and of the plant biostimulant Phylgreen^®^ on the metabolic profile of industrial tomato plants were investigated, combining the treatments with different types of mulching with commercial biodegradable films (Ecovio^®^ and MaterBi^®^). Previously, Di Mola and colleagues [15] demonstrated that in the same field experiment used in the present study, both biodegradable films and biostimulants determined a significant yield increase in processing tomatoes. Moreover, quality traits of fruits (e.g., total soluble solid-TSS content, fruit firmness, antioxidant activity, etc.) were significantly affected by treatments.

Here the metabolomic analysis carried out on tomato leaves and berries allowed the identification of over 60 compounds, most of which were significantly different among the various treatments. They primarily belonged to the class of flavonoids, phenols, and alkaloids and derivatives. Similarly, Mhlongo et al. [29] observed a change in the metabolic profile of tomato plants inoculated with PGPR (plant-growth-promoting rhizobacteria) with the accumulation of flavonoids, glycoalkaloids, benzoates, hydroxycinnamates, and aromatic amino acids due to the influence of the PGPR strains on secondary metabolism. The untargeted analysis revealed that both in berry and leaf samples the application of biostimulants resulted in a greater number of differential compounds whose intensity statistically decreased compared to the control group. Interestingly, the individual application of biostimulants showed greater differences compared to the combination of T22 and Phylgreen^®^. A recent study confirmed that the application of T22 caused a consistent change at both transcriptomic and metabolomic levels in tomato plants [30]. The authors found that treating tomato plants with the microorganism and then infesting them with the aphid *Macrosiphum euphorbiae* induced defence-related genes and accumulation of isoprenoids in leaves. Foliar applications of an extract based on *A. nodosum* on vines led to variations in plant physiology, grape quality and secondary metabolites present in the skins [31]. The presence of mulch films also determined changes in the metabolomes of the treated plants, with similar results between the two types of mulch in terms of intensity of differential metabolites. Interestingly, Di Mola et al. [15] found that both biodegradable films elicited a greater number of marketable tomato fruits compared to bare soil (control), possibly due to the increase in soil temperature, and elicited lipophilic antioxidant activity and a higher accumulation of ascorbic acid. The presence of biodegradable plastic films is known for its influence on the composition of the soil microflora [19,32,33]. When exposed to atmospheric agents, biodegradable mulch films lead to an enrichment of fungi, in particular Ascomycetes, and bacteria belonging to the phyla Proteobacteria, Actinobacteria, and Firmicutes in the soil [33]. The changes observed in the metabolomic profiles of the plants treated in the presence of biodegradable mulch films could be explained by the enrichment of soil microflora and by plant–rhizobiome interaction.

With a targeted approach, aimed at studying specific secondary metabolites, it was possible to evaluate the effects of treatments and mulch biofilms in terms of relative abundance variations. The analysis revealed that the trends of differentially accumulated metabolites in the examined tissues and organs when comparing various mulches or biostimulants, did not always involve the same compounds. Among the most representative compounds in the Solanaceae family, alkaloids are prominent, particularly in tomatoes, where α-tomatine, β1-tomatine, and tomatidine are found. These compounds are particularly abundant in green tomatoes, as well as in leaves and flowers [34]. α-Tomatine, the most abundant, is a glycoalkaloid structurally composed of tomatidine and β-lycotetraose, whose levels are mainly associated with the phenological stage of different organs and the genotype of the plant [35]. In tomato berries the levels of this glycoalkaloid are drastically reduced during ripening; in fact, unripe green tomatoes contain over 500 mg of α-tomatine per kg of fresh fruit, but during ripening, the compound is largely degraded and ripe tomatoes have approximately 5 mg per kg of fresh fruit [36]. Kozuke and colleagues found that 20 days after flowering, the concentration levels of dehydrotomatin and α-tomatine decreased by 96% and 94%, respectively, and that, by 50 days after flowering, the glycoalkaloid completely disappeared [34].

Certain compounds, such as dehydrotomatine, exhibited a strong increase in abundance in the berries of plants treated with biostimulants, and a decrease in the leaves of plants grown on mulched soil and treated with T22. On the other hand, compounds like α-tomatine showed differential accumulation only in some comparisons. This may be attributed to the fact that the metabolism of these alkaloids in fruits is regulated differently and independently from that of the vegetative parts [37].

Regarding tomatidine, significant increases in tomato berries were associated with the application of T22, both alone and in combination with Phylgreen^®^. A similar result was observed in tomatoes treated with different strains of *Trichoderma*, either alone or in combination with other microorganisms, as reported by Lanzuise et al. [38]. Like all glycoalkaloids, both α-tomatine and tomatidine have antifungal properties that play a role in plant defence mechanisms against insects and phytopathogenic bacteria [39]. These compounds exert their protective effects by targeting membrane sterols in pests and pathogens [38]. Hence, an increase in these compounds is certainly one of the advantages of applying treatments based on microbial biostimulants. The same microbial biostimulant was also found to increase when applied singly or in combination with *A. nodosum* formulation, the TSS content and the firmness of tomato fruits [15].

Di Mola and co-authors also found significant increases in colour parameters (L*: brightness) in tomato fruits treated with T22 both alone or in combination with Phylgreen^®^, but not in carotenoid content [15]. Here, the behaviour of compounds belonging to the class of flavonoids, a group of pigments found in plants and known for a broad spectrum of biological activities, was then evaluated. Flavonoids are a large family of low molecular weight polyphenolic compounds found in plant tissues, which include flavones, flavonols, flavonones, isoflavones and anthocyanins [40,41]. Of this class, the compound that exhibited a greater increase following treatment with a microbial biostimulant was (+)-medicarpin. Upon Phylgreen^®^ or T22 treatment, (+)-medicarpin increased by 60,000 and 80,000 times, respectively, in its abundance. This compound belongs to the group of phytoalexins, secondary metabolites produced by plants in response to natural or induced biotic stress [42,43,44,45]. In a 2021 study by Chavan and Koche, the medicarpin content was analysed in four chickpea cultivars, particularly in the cotyledons and in seedlings elicited with FCWE (*Fusarium* cell wall elicitor) [46]. The results demonstrated that high medicarpin content can be associated with greater resistance of the plant, as also supported by Kale and Choudhary [47], in a work carried out on peanuts.

Treatments based on the *A. nodosum* extract also led to an increase in phenolic and flavonoid compounds (quercetin 3-(2G-apiosylrutinoside), manghaslin and phaseolic acid), confirming the results of other studies [48] which demonstrated that foliar applications of *A. nodosum* led to a greater accumulation of anthocyanins and flavonols in grape skins, while a reduction in methoxylated anthocyanins, normally correlated to the presence of environmental stress, was observed [31]. These findings are consistent with the increase of up to 30% of total phenol content observed in tomato fruits treated with T22 and/or *A. nodosum*-based biostimulants compared to controls [15]. A similar increase in phenols and flavonoids was observed by Lola-Luz et al. [48]. Extracts of *A. nodsum* at different concentrations were applied to *Brassica oleaceae*, resulting in a greater accumulation of these substances, and leading to an improvement in the fitness of the plant.

These results confirm that both microbial and plant biostimulants can increase the production of several bioactive compounds, such as flavonoids, which play an important role in the defence of plant cells from pathogens, insects and UV rays [49]. Additionally, like most phenolic compounds, flavonoids protect the lipids within fruit pulp from oxidative stress [50], and act on the cell wall of pathogens by modifying their structure and reducing their enzymatic activity [51]. Flavonoids are also implicated in plant defence against microbial infections; studies by Mazzei et al. [52] and Pakora et al. [53], showed that flavonoids can inhibit spore germination and mycelium growth in soil-borne pathogens.

## 5. Conclusions

Our results demonstrated significant effects in qualitative aspects of industrial tomatoes when biostimulants were applied to plants grown on soil mulched with biodegradable films. These findings strengthen the results obtained by Di Mola et al., highlighting that the application of a biostimulant based on *Trichoderma* T22 or *A. nodosum* extract may improve the productive potentiality of processing tomato and enhance phenol content. 

Metabolomic analysis revealed that single biostimulant application led to similar variations in different plant tissues/organs, compared to combined treatment or control. These results indicate there was no synergism in the combined application of the two formulations for this specific crop. The presence of MaterBi^®^ biofilm resulted in an accumulation of differential compounds that significantly differed from both control and from plants mulched with Ecovio^®^ film. This suggests that cultivation with biodegradable mulch, in particular with MaterBi^®^ combined with a single treatment of either *T. afroharzianum* T22 or *A. nodosum*-based biostimulant yielded the best quality performance. Conversely, the combination of the two products did not seem to determine additional benefits, as also observed for other qualitative traits (e.g., TSS content, firmness, ascorbic acid content, etc.) [15]. These findings offer promising opportunities for the development of bioformulations based on microorganisms and natural extracts, as well as for the use of biodegradable mulching films in open-field crops such as tomatoes. Overall, the study highlights the potential of biostimulants and biodegradable mulch films in sustainable agriculture practices, offering ways to enhance crop productivity and quality while minimizing environmental impacts.

## Figures and Tables

**Figure 1 jof-10-00097-f001:**
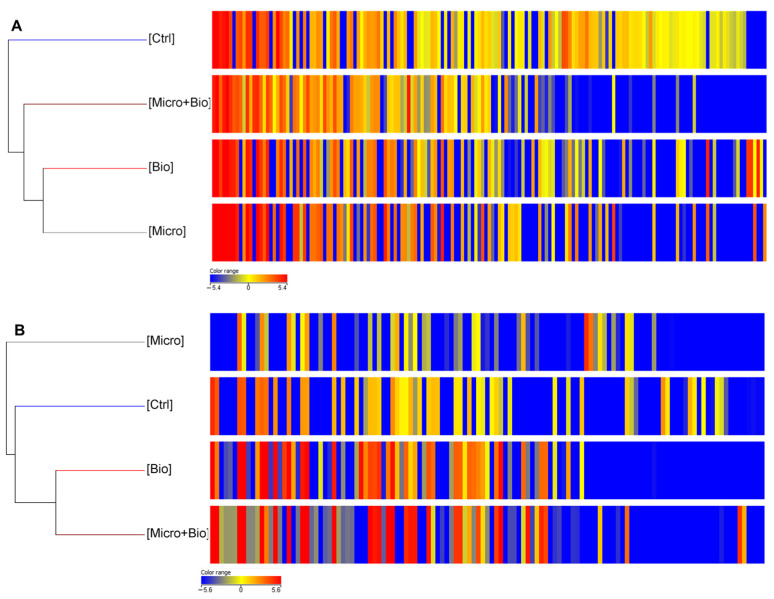
Heat maps and dendrograms obtained by comparing the differential metabolomic profiles of tomato leaves upon treatments with biostimulants, compared to plants treated with water only (Ctrl). The range of colours from blue to red shows how the intensities of the different compounds vary from the least abundant to the most abundant, respectively. (**A**) Differential metabolomic profiles obtained in positive ionization mode (ESI+). (**B**) Differential metabolomic profiles obtained in negative ionization mode (ESI−). Each column represents a metabolite, while data for each treatment is presented in rows. The heat map was developed using MassProfiler Professional bioinformatics software (Agilent Technologies) and statistical differences were determined by one-way ANOVA test (*p* < 0.05).

**Figure 2 jof-10-00097-f002:**
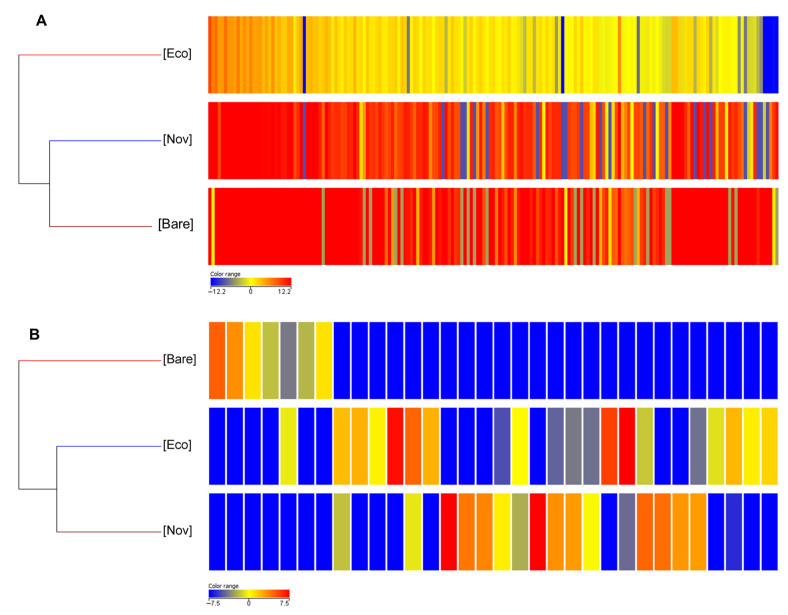
Heat map and dendrograms obtained by comparing the differential metabolomic profiles of tomato leaves cultivated on mulch biofilms and upon treatments with T22, compared to plants cultivated on bare soil and treated with T22. The range of colours from blue to red shows how the intensities of the different compounds vary from the least abundant to the most abundant, respectively. (**A**) Differential metabolomic profiles obtained in positive ionization mode (ESI+). (**B**) Differential metabolomic profiles obtained in negative ionization mode (ESI−). Each column represents a metabolite, while data for each treatment is presented in rows. The heat map was developed using MassProfiler Professional bioinformatics software (Agilent Technologies) and statistical differences were determined by one-way ANOVA test (*p* < 0.05).

**Figure 3 jof-10-00097-f003:**
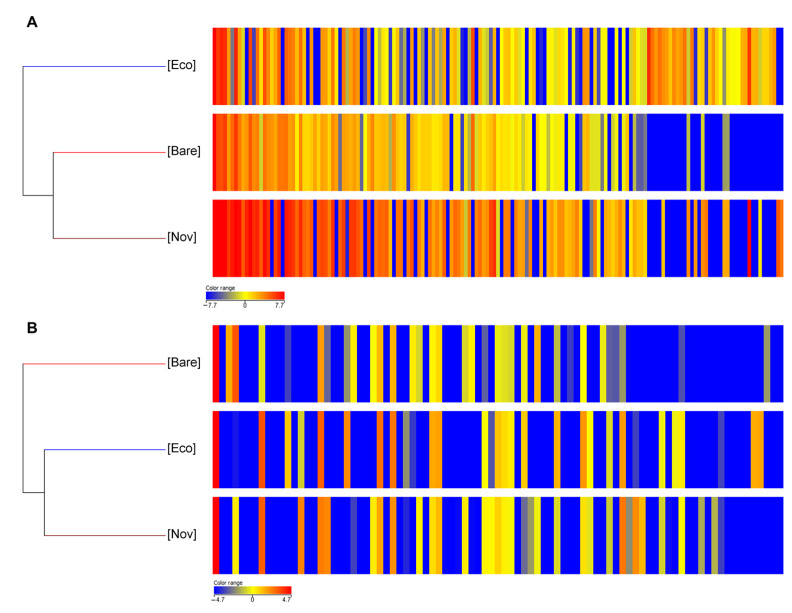
Heat maps and dendrograms obtained by comparing the differential metabolomic profiles of tomato leaves cultivated on mulch biofilms and upon treatments with the combination of both biostimulants, compared to plants cultivated on bare soil and treated with the combination of both biostimulants. The range of colours from blue to red shows how the intensities of the different compounds vary from the least abundant to the most abundant, respectively. (**A**) Differential metabolomic profiles obtained in positive ionization mode (ESI+). (**B**) Differential metabolomic profiles obtained in negative ionization mode (ESI−). Each column represents a metabolite, while data for each treatment is presented in rows. The heat map was developed using MassProfiler Professional bioinformatics software (Agilent Technologies) and statistical differences were determined by one-way ANOVA test (*p* < 0.05).

**Table 1 jof-10-00097-t001:** Putatively identified compounds in tomato extracts. Data include retention time (RT, expressed in minutes), compound name and class, chemical formula, experimental and theoretical monoisotopic mass, ionization mode, and type of tomato tissue (leaf, L; berry, B; both B/L).

N.	Origin	Ionization	RT	Compound	Class	Formula	**Experimental Mass (Da)**	**Theoretical Mass (Da)**
1	L	−	1.257	p-Coumaroylquinic acid	Phenol	C_16_H_18_O_8_	338.0854	338.10016753
2	L	+	1.786	Phenethylamine	Alkaloid	C_8_H_11_N	121.0895	121.089149355
3	L	−	1.831	Lupinisoflavone E	Flavonoid	C_25_H_26_O_7_	438.1655	438.16785316
4	L	−	4.578	4-Caffeoylquinic acid	Phenol	C_16_H_18_O_9_	354.0938	354.09508215
5	L	−	4.619	Kaempferol 3-glucosyl-(1->2)-galactoside-7-glucoside	Flavonoid glycoside	C_33_H_40_O_21_	772.2061	772.20620828
6	L	−	4.703	2′-Hydroxyisolupalbigenin	Flavonoid	C_25_H_26_O_6_	422.1707	422.17293854
7	L	−	4.705	Icariside B2	Glycoside	C_19_H_30_O_8_	386.1939	386.19406791
8	L	−	4.714	Manghaslin	Flavonoid glycoside	C_33_H_40_O_20_	756.2113	756.21129366
9	L	−	4.857	Quercetin 3-glucosyl-(1->6)-galactoside	Flavonoid glycoside	C_27_H_30_O_17_	626.148	626.14829948
10	L	+/−	4.895	Cucurbitacin K 2-O-beta-D-glucopyranoside	Steroidal glycoside	C_36_H_54_O_13_	694.3571	694.35644177
11	L	+	5.012	Hyperin	Flavonoid glycoside	C_21_H_19_O_12_	464.0943	464.09547607
12	L	+	5.020	Hydroxytomatine	Steroidal glycoalkaloid	C_50_H_85_NO_22_	1052.5536	1052.56260009
13	L	−	5.051	Phaseolic acid	Phenol	C_13_H_12_O_8_	296.0532	296.05321734
14	L	+	5.145	Fisetin	Flavonoid	C_15_H_10_O_6_	286.0457	286.04773803
15	L	+/−	5.146	Kaempferol 3-galactoside-7-rhamnoside	Flavonoid glycoside	C_27_H_30_O_15_	594.1581	594.15847025
16	L	−	5.146	Myricitrin	Flavonoid glycoside	C_21_H_20_O_12_	464.0954	464.09547607
17	L	−	5.192	Uttroside B	Saponin	C_56_H_94_O_28_	1214.5933	1214.59316234
18	L	+	5.228	Lycoperoside C	Steroidal saponin	C_52_H_85_NO_23_	1091.551	1091.55123796
19	L	+	5.355	β-2-Tomatine	Steroidal glycoalkaloid	C_44_H_73_NO_16_	871.4934	871.49293524
20	L	+/−	5.383	γ-Tomatine	Steroidal glycoalkaloid	C_39_H_65_NO_12_	739.4534	739.45067651
21	L	−	5.388	N-Malonyltryptophan	Aminoacid	C_14_H_14_N_2_O_5_	290.09	290.09027155
22	L	−	5.487	Acuminoside	Terpene glycoside	C_21_H_36_O_10_	448.2299	448.23084734
23	L	−	5.581	N-cis-Feruloyloctopamine	Phenol	C_18_H_19_NO_5_	329.1261	329.12632271
24	L	−	5.625	2-Methoxymedicarpin	Isoflavonoid	C_17_H_16_O_5_	300.1002	300.09977361
25	L	+	5.638	Colnelenic acid	Fatty acid	C_18_H_28_O_3_	292.2037	292.20384475
26	L	+	5.647	Apo-13-zeaxanthinone	Terpene	C_18_H_26_O_2_	274.1922	274.193280068
27	L	−	5.686	(+)-Medicarpin	Isoflavonoid	C_16_H_14_O_4_	270.0891	270.08920892
28	L	+/−	5.864	N-trans-Feruloyltyramine	Phenol	C_18_H_19_NO_4_	313.1315	313.13140809
29	B	+	1.342	Lotaustralin	Cyanogenic glycoside	C_11_H_19_NO_6_	261.1209	261.12123733
30	B	+	1.499	Linamarin	Cyanogenic glycoside	C_10_H_17_NO_6_	247.1054	247.10558726
31	B	+	1.551	Dihydrozeatin	6-alkylaminopurines	C_10_H_15_N_5_O	243.1199	221.12766012
32	B	+	1.788	(+)-Ligballinol	Furanoid lignan	C_18_H_18_O_4_	298.1235	298.12050905
33	B	+	1.799	Luteolin 3′,5-dimethyl ether	Flavonoid	C_23_H_24_O_11_	314.0776	314.07903816
34	B	−	4.136	5-Caffeoylquinic acid	Phenol	C_16_H_18_O_9_	354.095	354.09508215
35	B	−	4.206	L-Phenylalanine	Aminoacid	C_9_H_11_NO_2_	165.0789	165.078978594
36	B	+	4.743	Tryptamine	Alkylindole	C_10_H_12_N_2_	160.0993	160.100048391
37	B	−	4.939	Lycoperoside F	Steroidal glycoalkaloid	C_58_H_95_NO_29_	1269.5977	1269.59897599
38	B	−	5.026	Phloretin 3′,5′-Di-C-glucoside	Flavonoid glycoside	C_27_H_34_O_15_	598.1898	598.18977037
39	B	−	5.108	Taxifolin 3,7-dirhamnoside	Flavonoid glycoside	C_27_H_32_O_15_	596.1727	596.17412031
40	B	−	5.131	Naringenin-7-O-glucoside	Flavonoid glycoside	C_21_H_22_O_10_	434.1213	434.12129689
41	B	+	5.182	Anhydropisatin	Isoflavonoid	C_17_H_12_O_5_	296.0672	296.06847348
42	B	+	5.198	Indioside D	Saponin	C_51_H_84_O_23_	1064.5417	1064.54033892
43	B	+/−	5.226	Biochanin A	Flavonoid	C_16_H_12_O_5_	284.0681	284.06847348
44	B	−	5.254	12-Hydroxyjasmonic acid glucoside	Fatty acyl glycoside	C_18_H_28_O_9_	388.1723	388.17333247
45	B	+	5.278	Phloroglucinol	Benzenetriol	C_6_H_6_O_3_	126.0318	126.031694049
46	B	−	5.410	Formononetin 7-O-beta-D-glucoside-6″-O-malonate	Flavonoid	C_25_H_24_O_12_	516.1261	516.12677620
47	B	−	5.416	N-Acetyltryptophan	Aminoacid	C_13_H_14_N_2_O_3_	246.099	246.10044231
48	B	+	5.737	Isoesculeogenin A	Saponin	C_27_H_45_NO_4_	447.3326	447.33485892
49	B	−	6.030	(2R,3R)-fustin	Flavonoid	C_15_H_12_O_6_	288.0632	288.06338810
50	B	+	6.221	Butin	Flavonoid	C_15_H_12_O_5_	272.0677	272.06847348
51	B	−	7.207	Nordihydrocapsiate	Phenol	C_17_H_26_O_4_	294.1835	294.18310931
52	B	+/−	9.733	Colneleic acid	Fatty acid	C_18_H_30_O_3_	294.2197	294.21949481
53	B/L	−	4.796	Quinic acid	Cyclic carboxylic acid	C_7_H_12_O_6_	192.0631	192.06338810
54	B/L	+/−	4.851	Quercetin 3-(2G-apiosylrutinoside)	Flavonoid glycoside	C_32_H_38_O_20_	742.1959	742.19564360
55	B/L	+	5.009	Panasenoside	Flavonoid glycoside	C_27_H_30_O_16_	610.1529	610.15338487
56	B/L	+	5.010	Robinetin	Flavonoid	C_15_H_10_O_7_	302.0414	302.04265265
57	B/L	−	5.029	Isoorientin 2″-O-glucopyranoside	Flavonoid glycoside	C_27_H_30_O_16_	610.1535	610.15338487
58	B/L	−	5.093	Caffeic acid	Phenol	C_15_H_18_O_9_	180.0419	180.04225873
59	B/L	+/−	5.229	Dehydrotomatine	Steroidal glycoalkaloid	C_50_H_81_NO_21_	1031.5343	1031.53010859
60	B/L	+/−	5.267	α-Tomatine	Steroidal glycoalkaloid	C_50_H_83_NO_21_	1033.545	1033.54575866
61	B/L	+	5.273	Robeneoside B	Steroidal glycoalkaloid	C_45_H_73_NO_17_	899.488	899.48784986
62	B/L	+	5.303	δ-Tomatine	Steroidal glycoalkaloid	C_33_H_55_NO_7_	577.3983	577.39785309
63	B/L	+/−	5.318	β1-Tomatine	Steroidal glycoalkaloid	C_45_H_75_NO_17_	901.4983	901.50349992
64	B/L	+	6.975	Tomatidine	Steroidal alkaloid	C_27_H_45_NO	415.3428	415.345029678
65	B/L	+	7.213	Solasodine	Steroidal alkaloid	C_27_H_43_NO_2_	413.329	413.329379614

**Table 2 jof-10-00097-t002:** Number of differential metabolites whose intensity increased (UP) or decreased (DOWN) with respect to the control groups in tomato leaves following treatments with biostimulants. The values indicate the total number of compounds analyzed in positive (ESI+) and negative (ESI−) ionization modes. The plants, grown on bare soil (Bare), were inoculated with *Trichoderma* (Micro), the plant biostimulant Phylgreen^®^ (Bio) or with the combination of the two (Micro+Bio) and grown on bare soil (bare) or on mulch biofilms (Eco and Nov). Control: plants grown on bare soil and treated with water only (Bare); plants grown on bare soil and treated with *Trichoderma* (Bare Micro); plants grown on bare soil and treated with the combination of the two biostimulants (Bare Micro+Bio).

Source: Tomato Leaves	Bare Micro vs. Bare	Bare Bio vs. Bare	Bare Micro+Bio vs. Bare	Eco Micro vs. Bare Micro	Nov Micro vs. Bare Micro	Eco Micro+Bio vs. Bare Micro+Bio	Nov Micro+Bio vs. Bare Micro+Bio
UP	129	159	138	46	53	142	176
DOWN	159	129	150	169	162	104	70

**Table 3 jof-10-00097-t003:** Number of differential metabolites whose intensity increased (UP) or decreased (DOWN) compared to the control groups in tomato berries following treatments with biostimulants. The values indicate the total number of compounds analyzed in positive (ESI+) and negative (ESI−) ionization modes. The plants, grown on bare soil, were inoculated with *Trichoderma* (Micro), the plant biostimulant Phylgreen^®^ (Bio) or with the combination of the two (Micro+Bio). Control: plants grown on bare soil and treated with water only (Bare); plants grown on bare soil and treated with *Trichoderma* (Bare Micro); plants grown on bare soil and treated with the combination of the two biostimulants (Bare Micro+Bio).

Source: Tomato Berries	Bare Micro vs. Bare	Bare Bio vs. Bare	Bare Micro+Bio vs. Bare	Eco Micro vs. Bare Micro	Nov Micro vs. Bare Micro	Eco Micro+Bio vs. Bare Micro+Bio	Nov Micro+Bio vs. Bare Micro+Bio
UP	106	92	73	225	252	25	38
DOWN	208	222	241	129	102	52	39

**Table 4 jof-10-00097-t004:** Trend of the abundance of the main differential metabolites identified by LC-MS Q-TOF analysis in tomato leaf and berry extracts. For each compound, the trend in the samples treated with biostimulants was compared to the uninoculated samples obtained by plants grown in bare soil (BARE CTRL; on the left), or to the plant grown in bare soil but inoculated with T22 biostimulant (BARE MICRO; on the right). BIO, MICRO, BIO+MICRO = plants treated with plant (Phylgreen^®^) or microbial (T22) biostimulant or both, respectively. ECO, NOV = mulch biofilms Ecomont^®^ (ECO) or Materbi^®^, respectively.

Compound	vs. Bare Ctrl			vs. Bare Micro	
	Bio	Micro	Bio+Micro	Nov Micro	Eco Micro
p-Coumaroylquinic acid	↓	↓↓↓	↓	ND	ND
Phenethylamine	↓↓↓	↓↓↓	-	ND	ND
Lupinisoflavone E	↑	↑↑↑	-	↓↓↓	↓↓↓
4-Caffeoylquinic acid	-	↓↓↓	↑	↑↑↑	↑↑
Manghaslin	↑	-	↑↑↑	ND	ND
Cucurbitacin K 2-O-beta-D-glucopyranoside	↓	↓↓↓	↓	↑↑↑	↑↑↑
Hyperin	ND	ND	ND	↓↓↓	↓
Hydroxytomatine	-	-	↑↑↑	↓↓↓	↓
Phaseolic acid	↑	-	↑↑	ND	ND
Fisetin	-	↑	-	↑↑	↓
Myricitrin	ND	ND	ND	↑↑	↑↑↑
Uttroside B	↑↑	↓	↑↑	ND	ND
Lycoperoside C	ND	ND	ND	↓↓	↓
γ-Tomatine	-	-	↓	↑↑	↑↑↑
N-malonyltryptophan	↑	↓	-	ND	ND
2-Methoxymedicarpin	↑	↑↑	↑↑↑	ND	ND
Colnelenic acid	ND	ND	ND	↓↓↓	↓
Apo-13-zeaxanthinone	ND	ND	ND	↓↓↓	↓
(+)-Medicarpin	↑↑↑	↑↑↑	↑↑↑	ND	ND
N-trans-feruloyltyramine	ND	ND	ND	↓↓↓	↓
Lotaustralin	ND	ND	ND	↓	↓
Linamarin	↓↓↓	↓↓	↓	ND	ND
Dihydrozeatin	↓	-	↓↓	↓↓	↓↓
(+)-Ligballinol	-	↑	↓↓	ND	ND
Luteolin 3′,5′-dimethyl ether	↑	↑↑↑	-	↓↓↓	↓↓↓
5-Caffeoylquinic acid	↓	↓	↓	-	-
L-Phenylalanine	↓	↓	↓	-	-
Lycoperoside F	↓	↓	↓	ND	ND
Phloretin 3′, 5′- Di-C-glucoside	ND	ND	ND	-	-
Taxifolin 3,7-dirhamnoside	↓	↓	↓	-	-
Naringenin-7-O-glucoside	↓↓	↓↓	↓	-	↑
Anhydropisatin	↓	↑	↑	↓↓	↓
Biochanin A	↓↓	↓↓	↑	-	↑↑
12-Hydroxyjasmonic acid glucoside	↓	↓	↓	-	-
Formononetin 7-O-beta-D-glucoside-6″-O-malonate	↓↓	↓	↓	-	-
N-Acetyltryptophan	ND	ND	ND	↑	↑
Isoesculeogenin A	↑	-	-	ND	ND
(2R,3R)-fustin	↑↑	↑↑	↑↑	-	-
Butin	ND	ND	ND	↑	↑↑
Nordihydrocapsiate	ND	ND	ND	↓↓↓	↓↓
Colneleic acid	ND	ND	ND	-	-
Quinic acid	↓ ↑	↓ ↓↓↓	↓ ↑	↓↓ ↓↓↓	↓ ↓↓↓
Quercetin 3-(2G-apiosylrutinoside)	ND ↑	ND -	ND ↑	- ND	- ND
Panasenoside	ND -	ND ↑	ND -	- ↓↓↓	- ↓
Robinetin	- ↑	- ↑	- ↑	ND ↓	ND ↓
Isoorientin 2″-O-glucopyranoside	- -	- -	- -	- ND	↑ ND
Caffeic acid	↓ -	↓ -	↓ -	ND ↑↑↑	ND ↑↑↑
Dehydrotomatine	↑↑ -	↑↑↑ ↓	↑↑↑ -	ND ↓↓↓	ND ↓
α-Tomatine	- -	- ↓	- ↑	ND ↓↓	ND ↓
Robeneoside B	ND ND	ND ND	ND ND	↑ ↓↓↓	- ↓
δ-Tomatine	- -	- -	- -	ND ↓↓↓	ND ↓
β 1-Tomatine	ND ↓	ND ↓↓↓	ND ↓	- ↓↓↓	↑↑ ↓
Tomatidine	↑ -	↑↑ -	↑↑ -	↑↑ ↓↓↓	↑ ↓
Solasodine	↑ -	- -	- -	↑↑ ↓↓↓	↑ ↓

Legend: the arrows show the trend in logarithmic scale of the differential compounds identified in leaves (green arrow) or in berries (red arrow). ↓, ↓↓, ↓↓↓: the abundance of the compound compared to the control decreased by 2–5 times, 6–10 times, >10 times, respectively; ↑, ↑↑, ↑↑↑: the abundance of the compound compared to the control increased by 2–5 times, 6–10 times, >10 times, respectively; -: the abundance of the compound compared to the control has changed by a factor < 2. ND = compound not detected.

## Data Availability

Datasets generated for this study are available on request to the corresponding author.

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
