# Peer review of "Biodegradable Mulch Films and Bioformulations Based on Trichoderma sp. and Seaweed Extract Differentially Affect the Metabolome of Industrial Tomato Plants"

_jof, 2024, doi:10.3390/jof10020097_

Round 1

Reviewer 1 Report

Comments and Suggestions for Authors

The submitted work is focused on the important issue of mechanisms to improve the quality of agricultural products. The use of biostimulants is widespread and the application of metabolite profiling can provide important information on their effects. In the introduction, the authors argue the importance of their work quite well and introduce the reader to the problem. The proposed design of the experiment allows to analyze the specifics of the effect of biostimulants of different origin both singly and in combination. The obtained results and conclusions are of value for the development of new methods for the management of tomato traits.

The text needs some improvement. Most importantly, it is necessary to describe the methods of data treatment in more detail. It is desirable to specify which databases were used. It is important to describe what data preprocessing manipulations were done (normalization, outlier detection...).  It is probably worth doing post-hoc tests for ANOVA. The method of clustering needs to be specified.

Lines 269-270 “…the combined treatment (T22 and Phylgreen®, MicroBio) cluster together with control group (Figure 1A) …”  but in Figure 1A the control variant is opposed to the branch containing the treated ones including "micro+bio".

In general, the work makes a good impression and provides new data on the issues under study The work can be published after a minor adjustment.

Author Response

Thank you very much for taking the time to review this manuscript. Please find the detailed responses below and the corresponding revisions in track changes in the re-submitted files.

Point-by-point response to Comments and Suggestions for Authors

Comments 1: The submitted work is focused on the important issue of mechanisms to improve the quality of agricultural products. The use of biostimulants is widespread and the application of metabolite profiling can provide important information on their effects. In the introduction, the authors argue the importance of their work quite well and introduce the reader to the problem. The proposed design of the experiment allows to analyze the specifics of the effect of biostimulants of different origin both singly and in combination. The obtained results and conclusions are of value for the development of new methods for the management of tomato traits.

Response 1: We thank the Reviewer for the positive comments.

Comments 2: The text needs some improvement. Most importantly, it is necessary to describe the methods of data treatment in more detail. It is desirable to specify which databases were used. It is important to describe what data preprocessing manipulations were done (normalization, outlier detection...).  It is probably worth doing post-hoc tests for ANOVA. The method of clustering needs to be specified.

Response 2: We agree with this comment. The text has been modified according to the Reviewer suggestions (see Lines 189-208). In particular, the methods of data treatment were better described, including the information about the database, data processing manipulations, and method of clustering.

Comments 3: Lines 269-270 “…the combined treatment (T22 and Phylgreen®, MicroBio) cluster together with control group (Figure 1A) …”  but in Figure 1A the control variant is opposed to the branch containing the treated ones including "micro+bio".

Response 3: Thank you for pointing this out. We modified the text to remove the inconsistency (see Lines 282-285).

Comments 4: In general, the work makes a good impression and provides new data on the issues under study The work can be published after a minor adjustment.

Response 4: We thank the Reviewer for the positive comment.

Reviewer 2 Report

Comments and Suggestions for Authors

The paper describes a study on the influence of bio-based mulch used with Trichoderma sp. stimulating agents on the metabolome changes of industrial tomatoes. The analyses included both leaves and berries. The manuscript is well-written and provides some new data, extending the knowledge on the subject. I have only some minor suggestions for improvement, that the Authors could take into account while preparing a revised version.

Introduction is concise and clear. The section gives enough background information to lead the reader into the subject of the study. I have no major complaints regarding this part.

Materials and Methods were described extensively and precisely, definitely allowing for repeating the experiments. Methodology was adequately chosen to achieve the goals of the study.

The description of results is a bit dull and lacks the highlights that perhaps should be emphasized here. Nevertheless, the obtained results were presented clearly and the description is convincing. Font size in the heat map figures could be larger. Also, the description of the figures could be more extensive as it is not clear which classes or compounds were represented in the charts. Table numbering should be checked (is there Table 8 in the manuscript?).

Discussion: some information could be moved to Introduction or simply removed because it repeats basic knowledge and does not focus on the findings of the study. It may be a little distracting for the reader. Apart from that, the results were properly discussed and appropriate literature data supported them well. I do not have any particular complaints regarding this section.

Author Response

Thank you very much for taking the time to review this manuscript. Please find the detailed responses below and the corresponding revisions in track changes in the re-submitted files.

Point-by-point response to Comments and Suggestions for Authors

Comments 1: The paper describes a study on the influence of bio-based mulch used with Trichoderma sp. stimulating agents on the metabolome changes of industrial tomatoes. The analyses included both leaves and berries. The manuscript is well-written and provides some new data, extending the knowledge on the subject. I have only some minor suggestions for improvement, that the Authors could take into account while preparing a revised version.

Introduction is concise and clear. The section gives enough background information to lead the reader into the subject of the study. I have no major complaints regarding this part.

Materials and Methods were described extensively and precisely, definitely allowing for repeating the experiments. Methodology was adequately chosen to achieve the goals of the study.

Response 1: We thank the Reviewer for the positive comments.

Comments 2: The description of results is a bit dull and lacks the highlights that perhaps should be emphasized here. Nevertheless, the obtained results were presented clearly and the description is convincing. Font size in the heat map figures could be larger. Also, the description of the figures could be more extensive as it is not clear which classes or compounds were represented in the charts. Table numbering should be checked (is there Table 8 in the manuscript?).

Response 2: The text has been modified according to the Reviewer suggestions. In particular, we increased the font size in the heat map figures and revised table numbering. About the description of the Figures 1-3 and S2-S4, the charts showed the metabolomic profiles of leaves and berries, respectively, using an untargeted approach. Therefore, no identified compounds were indicated. Subsequently, among the 65 identified compounds (Table 1), the abundance of the main differential metabolites has been reported in Table 4 and was evaluated according to the treatment applied (Fig. S5). In the latter case, the abundance refers to single identified compounds, that were listed on the right side of each heat map.

Comments 3: Discussion: some information could be moved to Introduction or simply removed because it repeats basic knowledge and does not focus on the findings of the study. It may be a little distracting for the reader. Apart from that, the results were properly discussed and appropriate literature data supported them well. I do not have any particular complaints regarding this section.

Response 3: Repetitions were removed from Discussion section and some information were moved to Introduction. We thank the Reviewer for the positive comments.

Reviewer 3 Report

Comments and Suggestions for Authors

The author of the manuscript jof-2809596 describes the effects of two commercial biostimulants, applied single or combined with two biodegradable mulch films, on the metabolome of industrial  (field-grown) tomato plants. The manuscript needs significant improvements before being considered for publication.

My main concern is related to the reason for splitting the data from a single research idea – the influence of plant biostimulants and biodegradable mulch films on field-grown tomato plants. In the Introduction and Discussion sections, the authors must explain the reason for presenting the influence of plant biostimulants and biodegradable mulch films on the yield and quality of the field-grown tomato and metabolome in two separate works, the present manuscript and Di Mola et al. (2023), Agronomy13(3), 901. The experiment is not very complex – the field experiment was done one year in a single location. The jof-2809596 manuscript, combined with the published paper based on the same field experiment, will not exceed 25 pages. Therefore, the resulting work is not very difficult to read (and MDPI has no page limit for a submitted manuscript). The methodology is complementary and not different. No different hypotheses are tested in these two works (based on the same field experiment, differently presented). Both works have similar conclusions - combining plant biostimulants and biodegradable mulch film increases fruit quality. In the combined form, more valuable conclusions could result – for example, the relationship between total soluble solids from tomato fruits and the metabolomic data.

There are contradictions in the presentation of the field experiment in these two works, the jof-2809596 manuscript and Di Mola et al. (2023) paper. The surface of a (sub)plot is 3 m2 (Di Mola et al. 2023) and 3.5 x 0.70 m (2.45 m2) in the present manuscript. The seaweed extract has an ”application dose of 2 L ha-1 diluted in 30 L of water” (current manuscript) and3 mL L−1 — Di Mola et al. (2023) paper.

The metabolomic approach is limited to presenting the heatmaps without further interpretation and discussion of these findings. Nowadays, based on metabolomic data, predictions of the metabolic pathway are possible (Toubiana et al. (2019). Combined network analysis and machine learning allows the prediction of metabolic pathways from tomato metabolomics data. Communications biology, 2(1), 214). Such predictions, combined with the active ingredients from Ascophyllum nodosum extract and active metabolites of Trichoderma afroharzianum T22 strains and their expected mechanism of action, could contribute to a better understanding of plant biostimulants mechanism of action.

Author Response

Thank you very much for taking the time to review this manuscript. Please find the detailed responses below and the corresponding revisions in track changes in the re-submitted files.

Point-by-point response to Comments and Suggestions for Authors

Comments 1: The author of the manuscript jof-2809596 describes the effects of two commercial biostimulants, applied single or combined with two biodegradable mulch films, on the metabolome of industrial  (field-grown) tomato plants. The manuscript needs significant improvements before being considered for publication.

My main concern is related to the reason for splitting the data from a single research idea – the influence of plant biostimulants and biodegradable mulch films on field-grown tomato plants. In the Introduction and Discussion sections, the authors must explain the reason for presenting the influence of plant biostimulants and biodegradable mulch films on the yield and quality of the field-grown tomato and metabolome in two separate works, the present manuscript and Di Mola et al. (2023), Agronomy, 13(3), 901. The experiment is not very complex – the field experiment was done one year in a single location. The jof-2809596 manuscript, combined with the published paper based on the same field experiment, will not exceed 25 pages. Therefore, the resulting work is not very difficult to read (and MDPI has no page limit for a submitted manuscript). The methodology is complementary and not different. No different hypotheses are tested in these two works (based on the same field experiment, differently presented). Both works have similar conclusions - combining plant biostimulants and biodegradable mulch film increases fruit quality. In the combined form, more valuable conclusions could result – for example, the relationship between total soluble solids from tomato fruits and the metabolomic data.

Response 1: As correctly noted by the Reviewer, the same field trial is the focus of manuscript jof-2809596 and Di Mola et al. 2023  (as is also indicated in the manuscript jof-2809596). Although using the same experiment, the two manuscripts differ substantially in the objectives of the research investigation regarding the treatments and data.

In Di Mola et al. 2023, the authors “aimed to assess the agronomic response and qualitative traits of fruits of processing tomatoes grown on different biodegradable mulching films and treated with biostimulants based on Trichoderma sp. and/or Ascophyllum nodosum”. Therefore, overall yield and quality parameters were analyzed on the harvested fruits relative to the mulching biofilm and/or biostimulant treatments used in the cultivation of tomato.

In the present manuscript, the focus is on the metabolomic response observed in both tomato leaves and berries following the single or combined applications of the two biostimulants and the two different biodegradable mulch films. The metabolomic analysis provides a broader picture of the treated plant responses reflected in the biosynthetic pathways, that is to say, the cause and effects of cultivation practices on vegetative tissues.

We believe that a combination of all these diverse data in a single manuscript would have made it difficult to conceptually explain, interpret the results and produce a consolidated logical discussion. For instance, it is not possible to directly correlate total soluble solids (that in tomato fruits takes into account mainly sugar content, e.g. fructose) to alkaloid or flavonoid accumulation.

Comments 2: There are contradictions in the presentation of the field experiment in these two works, the jof-2809596 manuscript and Di Mola et al. (2023) paper. The surface of a (sub)plot is 3 m2 (Di Mola et al. 2023) and 3.5 x 0.70 m (2.45 m2) in the present manuscript. The seaweed extract has an ”application dose of 2 L ha-1 diluted in 30 L of water” (current manuscript) and “3 mL L−1”  — Di Mola et al. (2023) paper.

Response 2: Agreed. The details of the methods were erroneously described. We have accordingly, corrected the description, and it now corresponds to the Di Mola et al. (2023) paper. Thank you for noting the discrepancy.

Comments 3: The metabolomic approach is limited to presenting the heatmaps without further interpretation and discussion of these findings. Nowadays, based on metabolomic data, predictions of the metabolic pathway are possible (Toubiana et al. (2019). Combined network analysis and machine learning allows the prediction of metabolic pathways from tomato metabolomics data. Communications biology, 2(1), 214). Such predictions, combined with the active ingredients from Ascophyllum nodosum extract and active metabolites of Trichoderma afroharzianum T22 strains and their expected mechanism of action, could contribute to a better understanding of plant biostimulants mechanism of action.

Response 3: We thank the Reviewer for the comment. We agree that predictions of the metabolic pathway could be essential for comprehending functional metabolomics. However, understanding the expected mechanism of action of plant biostimulants was beyond the scope of the present study. Here we aimed to discriminate the effects of field treatments in terms of accumulated differential metabolites in tomato leaves and berries, and we discuss the effects of single or combined biostimulant applications relative to the cultivation with the two mulching biofilms. Our results provide support for the development of novel management practices based on the integrated use of plant and microbial biostimulants, thus highlighting the potential of biostimulants and biodegradable mulch film applications in sustainable agriculture.

Round 2

Reviewer 3 Report

Comments and Suggestions for Authors

I asked the authors of manuscript jof-2809596 to present in the Introduction and Discussion sections the reason for splitting the work related to the influence of plant biostimulants and biodegradable mulch films on the yield and quality of the field-grown tomato and metabolome in two separate papers, the present manuscript and Di Mola et al. (2023), Agronomy, 13(3), 901. They did not respond to my request. They just argued in their response to me that “the two manuscripts differ substantially in the objectives of the research investigation regarding the treatments and data”. However, the issue was not related to the similar data. The issue is that we have: (i) a single research idea – the influence of plant biostimulants and biodegradable mulch films on field-grown tomato plants; (ii) the same general objective  — “the development of novel management practices based on the integrated use of plant and microbial biostimulants, thus highlighting the potential of biostimulants and biodegradable mulch film applications in sustainable agriculture”; (iii) the same field experiment  - done one year, in one location. The present manuscript has only a different method – metabolomics, limited to findings. The readers must understand the reason to split a paper telling the whole story into two publications. I maintain my position – the arguments must be presented in the manuscript at the end of Introduction section, after L103. The benefits of metabolomics for developing novel management practices must be discussed in direct relationship with the findings from the previous work. The advantages of metabolomics findings compared to the previous paper must be highlighted.

Author Response

Thank you very much for taking the time to review this manuscript. Please find the detailed responses below and the corresponding revisions in track changes and yellow highlighter in the re-submitted files.

Point-by-point response to Comments and Suggestions for Authors

Comments 1: I asked the authors of manuscript jof-2809596 to present in the Introduction and Discussion sections the reason for splitting the work related to the influence of plant biostimulants and biodegradable mulch films on the yield and quality of the field-grown tomato and metabolome in two separate papers, the present manuscript and Di Mola et al. (2023), Agronomy, 13(3), 901. They did not respond to my request. They just argued in their response to me that “the two manuscripts differ substantially in the objectives of the research investigation regarding the treatments and data”. However, the issue was not related to the similar data. The issue is that we have:

(i)        a single research idea – the influence of plant biostimulants and biodegradable mulch films on field-grown tomato plants;

(ii)       the same general objective  — “the development of novel management practices based on the integrated use of plant and microbial biostimulants, thus highlighting the potential of biostimulants and biodegradable mulch film applications in sustainable agriculture”;

(iii)      the same field experiment  - done one year, in one location. The present manuscript has only a different method – metabolomics, limited to findings.

The readers must understand the reason to split a paper telling the whole story into two publications. I maintain my position – the arguments must be presented in the manuscript at the end of Introduction section, after L103. The benefits of metabolomics for developing novel management practices must be discussed in direct relationship with the findings from the previous work. The advantages of metabolomics findings compared to the previous paper must be highlighted.

Response 1: We have modified the manuscript according to the Reviewer comments (see Lines 104-109, 479-480, 503-506, 544-549, 568-570, 586-588, 598-599). In particular, at the end of Introduction section, the reason for examining metabolomic data separately from those reported in Di Mola et al. has been included. In Discussion and Conclusions sections results obtained from metabolomic analysis were compared to the previous paper.

Round 3

Reviewer 3 Report

Comments and Suggestions for Authors

The authors made the requested modifications.